# Compositional Dependence of Pore Structure, Strengthand Freezing-Thawing Resistance of Metakaolin-Based Geopolymers

**DOI:** 10.3390/ma13132973

**Published:** 2020-07-03

**Authors:** Dongming Yan, Lingjun Xie, Xiaoqian Qian, Shaoqin Ruan, Qiang Zeng

**Affiliations:** College of Civil Engineering and Architecture, Zhejiang University, Hangzhou 310058, China; dmyan@zju.edu.cn (D.Y.); 21512179@zju.edu.cn (L.X.); qianxq1@zju.edu.cn (X.Q.); sruan001@zju.edu.cn (S.R.)

**Keywords:** geopolymer, alkaline activator material, pore structure, freezing-thawing

## Abstract

The understanding of the composition dependent properties and freezing-thawing (F-T) resistance of geopolymer materials is vital to their applications in cold regions. In this study, metakaolin-based geopolymer (MKG) mortars were fabricated by controlling the Si/Al ratio and the Na/Al ratio. The pore structure and strength were measured by mercury intrusion porosimetry and compression tests, respectively, which both showed obvious correlations with the material composition. Mass loss, strength loss, visual rate, and microscopic observation were adopted to assess the changes of the material properties and microstructure caused by F-T loads. The results showed that the strength-porosity relationship roughly followed a linear plot. Increases of the Si/Al ratio increased the capillary pore volume, but decreased the gel pore volume and the F-T resistance. Increases of the Na/Al ratio decreased the gel pore, but roughly enhanced the F-T resistance. The MKG mortar at the Na/Al ratio of 1.26 showed the lowest total pore volume and the best F-T resistance. The mechanisms of our experimental observations were that the abundantly distributed air voids connected by the capillary pores facilitated the relaxation of hydraulic pressures induced by the freezing of the pore liquid. The findings of this work help better clarify the compositional dependence of the pore structure, strength, and freezing-thawing resistance of MKG materials and provide fundamental bases for their engineering applications in cold regions.

## 1. Introduction

The great amount of CO${}_{2}$ emissions from the production of ordinary Portland cement (OPC), around 0.8 kg CO${}_{2}$ per kg OPC [1], provides significant engineering, social-economic, and ecologic incentives to replace OPC with green cementitious materials with low CO${}_{2}$ emissions. Geopolymer, a type of inorganic alumino-silicate polymer synthesized from predominantly silicon (Si) and aluminum (Al) materials of geologically active minerals, may be a preferable binding material that can glue aggregates to fabricate concrete [2,3]. Most importantly, the CO${}_{2}$ emissions for producing one ton of geopolymer are only 20% of those for OPC [3]. The economical and environmental benefits would be further raised when geopolymers are synthesized with solid wastes with high value added applications [4,5,6,7,8,9]. The cementing or geopolymerization process of geopolymer precursors with alkaline activators can rapidly proceed to form a three-dimensional polymeric structure with complex Al-O-Si bonds and to generate continual material skeletons that enable high strengths [10]. With those advantages, geopolymer-based materials possess a series of excellent performances with promising potentials in engineering applications, including high mechanical strength, great fire resistance, enhanced leaching resistance, and improved contaminate stabilization [11,12,13,14,15].

The engineering performances of geopolymer materials are highly dependent on the material composition, synthesizing method, and curing scheme [10,11,16,17,18,19]. Among these factors, the material composition may play the most important role, because it decisively impacts the material structures, mechanical properties, and durability performances. For example, the Si/Al ratio and the Na(K)/Al ratio are generally elaborately designed to promote the mechanical properties of geopolymer materials. Rowles and Connor [11] found that metakaolin-based geopolymers (MKG) at the Si/Al ratios of 2–2.5 and Na/Al ratios of 1–1.5 showed optimized strengths. Using extensive experimental tests and statistic analyses, Lahoti et al. [17] reported that the Si/Al ratios between 1.5 and 2.2 and the Na/Al ratios between 0.8 and 1.5 enabled the fabrication of high strength MKG mortars over 60 MPa. Composition designs not only impact the material strength, but also alter the pore structure of MKG materials. For example, Duxon et al. [20] tested the pore size distribution (PSD) of MKG materials at different Si/Al ratios and found that the rise of the Si/Al ratio would narrow the pore sizes at the nanoscale. Nevertheless, these pore structure alterations may lead to different responses of geopolymer materials to freezing-thawing (F-T) cycles, which has not been clarified in previous studies.

The durability of geopolymer materials against F-T loads faces great challenges when geopolymer-based structures are in cold regions especially for hydraulic engineering applications, such as dams, aqueducts, and docks. One important challenge is that the pore structure of geopolymer materials varies with the material composition [20]. Therefore, the ordinary tests on the mechanical performances of geopolymer materials subject to F-T loads are far from approaching the real F-T degradation in the materials [21,22,23].

As an important durability index, F-T resistance has been widely tested in ordinary structure materials such as concrete, mortar, and masonry [24,25,26]. For all types of porous materials, the F-T damage mechanisms are similar. Generally, F-T damages occur when the pore pressures built from the transition of pore water to ice exceed the material strength [27,28,29,30,31]. Generally, the freezing pore pressures consist of the hydraulic pressures caused by the water-ice density differences and the crystallization pressures induced by the free energy differences [32,33]. The pore confinement will greatly influence the freezing process of water [28,34,35], which, consequently, impacts the frost damages of porous materials. Therefore, it is difficult to explore the F-T resistances of porous materials with different pore structures.

In the present work, we aimed to deepen the understandings of correlations between the pore structure and F-T resistance of MKG materials. MKG mortars with different pore structures were synthesized by controlling the Si/Al ratio and the Na/Al ratio. The pore structure of the MKG mortars was measured by mercury intrusion porosimetry (MIP), and its correlations to the compressive strengths were evaluated. The F-T performances of the MKG mortars were assessed and linked to the pore structure. The results of this study shed much light on the compositional dependence of the pore structure, strength, and freezing-thawing resistance of MKG materials.

## 2. Experiments

### 2.1. Materials and Specimen Preparation

A commercial metakaolin powder (Metamax, Basf Co. LTD., Shanghai, China) was used as the alumino-silicate source of the geopolymer materials. Using laser particle size analysis (LS-230, Coulter), the mean particle size was measured as 5.91 μm, and the 90%-passed particle size was 13.59 μm. The particle size distribution of the metakaolin powder is shown in Figure 1a. The packing density of the metakaolin powder was 0.422 g/mL. The chemical composition of the metakaolin powder was measured by X-ray fluorescence (XRF) spectrometer analysis (SHIMADU XRF-1800, Shimadzu Global Laboratory Consumables Co., Ltd. Shanghai, China). ${\mathrm{SiO}}_{2}$ and ${\mathrm{Al}}_{2}O_{3}$ occupied 57.47 wt.% and 39.81 wt.% of the total oxides, respectively (Table 1). The XRD patterns of the metakaolin are shown in Figure 1b. Only amorphous halo and anatase are observed in the figure, suggesting that the metakaolin contained the amorphous ${\mathrm{SiO}}_{2}$ and ${\mathrm{Al}}_{2}O_{3}$ with highly reactive potentials.

The alkaline activator used in this work was prepared by liquid sodium silicate (SiO${}_{2}$ = 27.35%, Na${}_{2}$O = 8.42%, and H${}_{2}$O = 64.23%) and pellet sodium hydroxide (Na${}_{2}$O = 77.4%, H${}_{2}$O = 22.5%, and impurity = 0.1%). Fine sands with a fineness modulus of 1.75 and a density of 2510 kg/m${}^{3}$ were adopted as the fine aggregates for preparing the MKG mortars.

In order to investigate the possible influences of material composition (Si/Al ratio and Na/Al ratio) on the pore structure and mechanical properties of MKG materials, two mortar groups were designed. For the first group (MKG-1 to MKG-3), the Si/Al ratios varied from 2.01 to 2.62 with the Na/Al ratio fixed at 1.01. For the second group (MKG-3 to MKG-5), the Si/Al ratio was fixed at 2.62, while the Na/Al ratio increased from 1.01 to 1.36. The mix proportions and nomenclature are shown in Table 2. In all mixes, a water-to-binder (w/b) ratio of 0.62 and a sand-to-binder (s/b) ratio of 3 were used, so that the influences of w/b and s/b could be eliminated. The alkaline activators were first synthesized according to the stoichiometric balances required by the different Si/Al and Na/Al ratios. After settling down for 12 h, the activator solutions were readily prepared for the geopolymer synthesis.

When preparing the fresh MKG mortars, a readily settled alkali activator solution was first poured into a Hobart mixer bowl, followed by the precisely weighed metakaolin powder and water. A low-speed stirring for 3 min was conducted to obtain the MKG paste slurries. After that, sands were added into the paste slurries with another 3 min stirring to homogenize the MKG mortar. The mortar slurries were then cast into cuboid molds with dimensions of 40×40×160 mm${}^{3}$. High-frequency vibrations were conducted on the fully filled molds to further remove the air bubbles entrapped in the mortars. The specimens, together with the molds, were covered with plastic film to avoid any loss of water that may cause microstructure alterations. They later experienced a standard curing (20 ± 1 °C and 90% relative humidity) for 2 d to increase the strengths. After demolding, all the mortar specimens were stored in a curing chamber to 28 d.

### 2.2. Testing Methods

#### 2.2.1. Strength

Compressive strengths were tested in a TYE-300D (Jianyi experiment instrument Co. LTD., WuXi, Jiangsu, China) automatic mechanical testing machine with a loading speed of 2.4 kN/s. Six specimens of each mix were tested and averaged to guarantee the data reliability and productivity.

#### 2.2.2. Freezing-Thawing

F-T durability is important when MKG materials are applied in hydraulic engineering structures in cold regions. F-T tests were performed using a CDR-5 rapid freeze-thaw machine according to the Test Code for Hydraulic Concrete [36]. Before the F-T tests, all the specimens were immersed in tap water for 4 days to enhance the water saturation degrees. The mass of those water-filled specimens was weighed by a highly accurate balance and termed as the initial mass $m_{0}$. Later, the specimens were loaded in water-filled vessels with the water surface 20 mm higher than the top surfaces of the specimens. Those vessels that contained the MKG specimens and water were then removed into the F-T testing machine. The temperature was controlled between −17 and 8 °C. A freezing (thawing) course took 1.5–2.5 h (1–1.5 h) to decrease (raise) the temperature. One complete F-T course cost 2.5–4 h.

After 50 F-T cycles, the specimens were removed from the freezing chamber, and the excess surface water was cleaned by absorbent papers. The mass of the surface saturated specimens was measured and termed as $m_{FT}$. The relative mass loss Δm was evaluated as:(1)$$\Delta m=\frac{m_{0}-m_{FT}}{m_{0}}$$

When the specimens were completely damaged after the F-T loads, neither the strength nor the mass was measured.

#### 2.2.3. Pore Structure

The central part of each MKG mortar was crushed into small MKG particles (around 10 mm in diameter) for MIP tests. Owing to the reproductive data and broad pore ranges of MIP [37], only one MIP test for each mortar was conducted. MIP tests were carried out using an Autopore IV 9510 (Micromeritics Instrument Corp., Norcross, GA, USA). The applied intrusion pressures were set from 1.4 kPa to 207 MPa with the equilibrium time of 10 s for each pressure step. Mercury fronts invade pores or cavities when the exerted forces are high enough to overcome the barriers induced by the surface forces on the pore curvatures. By accurately recording the mercury intrusion (extrusion) volume (or mass) at each pressurization step, the pore structure of a porous material can be assessed. Generally, the Washburn equation [38] is used to link the applied pressures (*P*) to the pore sizes *D*, i.e., D=−4γcosθ/P, where γ is the surface tension of mercury and θ is the contact angle between mercury and the pore wall. Note that the Washburn equation requires the gradually sized and connected pore system in cylindrical geometry. By taking the commonly used physical parameters of mercury, i.e., the contact angle of 130° and the surface tension of 485 N/m [39], the minimum and maximum accessible pore sizes, according to the Washburn equation, were estimated as 6 nm and 360 μm, corresponding to the maximum and minimum applied pressures, respectively.

#### 2.2.4. Micro Morphology

Scanning electron microscopy (SEM) in back-scattered electron (BSE) mode was tested in an equipment of Quanta FEG 650 (Thermo Fisher Scientific, Beijing, China) to observe the microstructure alterations of the MKG mortars induced by F-T cycles. BSE images can more clearly distinguish the pore phase (including the cracks) from the solid skeletons, because they have different electron back-scattering coefficients. The MKG mortar particles before and after F-T loads were oven-dried at 40 °C for 24 h and impregnated with a quick-hardening epoxy resin. Later, the solidified resin-covered samples were ground and polished by various grades of sand papers and diamond suspensions. After a short carbon coating, the smooth and flat samples were loaded in a sample platform for the SEM/BSE tests. The voltage of 30 kV and spot size of 5.0 nm were used to acquire the BSE images.

## 3. Materials’ Properties and Pore Structure

### 3.1. Pore Structure

Table 3 summarizes some characteristic pore parameters of the MKG mortars from MIP tests, i.e., the total porosity $\phi_{T}$, average pore size ($D_{a}=4V/A$), specific surface area (*A*), and threshold pore size ($D_{t}$). Obviously, mortars with different mix proportions showed different characteristic pore parameters. Similar trends were found between the total porosity and specific surface area and between the average pore size and threshold pore size. They all coarsely reflected the pore structure features of a porous material. A higher total porosity (or specific surface area) generally implies a more porous structure, while a larger average pore size suggests a coarser pore structure. Threshold pore sizes measure the connected throats formed from the interparticle continuum [40], which is generally a featured index of permeability. More specific discussions about the pore data with the material compositions are given below.

Figure 2 shows the MIP pore data of the MKG mortars, where the pore size distributions (PSDs) in the accumulative (APSD) and differential forms (DPSD) were acquired. While the MKG mortars showed different PSD shapes, similar physical interactions took place between the mercury fronts and pores. At the beginning of the MIP test, mercury first invaded the open cracks, gaps, cavities, and irregularities on the sample surfaces, which was termed as the surface-conformance effect [39]. This surface-conformance effect may be inevitable, because sample pretreatments such as cutting and drying [41] would always enhance the surface roughness. The rapid fillings of mercury at low pressures accounted for the rises in APSDs (Figure 2a) and the peaks in DPSDs at 100 μm (Figure 2b). Analysis showed that the mercury volumes for covering the surface roughness were all around 0.01 mL/g (Figure 2c), occupying 12–17% of the total intrusion volumes (Figure 2d). The results may show that the MKG samples had similar surface roughness.

As the applied pressure increased, mercury fronts penetrated into the thinner pores. Rapid mercury rises were observed between 50 and 10,000 nm (Figure 2a), which can be cataloged as the capillary pores. They were the connected spaces among the metakaolin particles, hydration products, and fine aggregates (known as the inter-particle spaces). The invasion of mercury into those inter-particle spaces led to the intrusion peaks between 100 and 2000 nm (Figure 2b and Table 3). It is noteworthy that MIP PSD always underestimates the pore sizes due to the “ink-bottle” effect [42]. Nevertheless, it measures the pore throats that establish the pore connections [40]. Pore analysis showed that the capillary pores between 50 nm and 10,000 nm occupied different volumes and ratios for different MKG mortars (Figure 2c,d). Specifically, more than 70 percent of the pores in the MKG-4 and MKG-5 samples were capillary pores (Figure 2d).

Under higher pressures, mercury could invade the pores below 50 nm (gel pores). Great PSD differences between the MKG mortars were found without uniform trends (Figure 2a,b). The MKG-1 sample had coarser PSDs and higher gel pore volumes than MKG-2 and MKG-3, while the MKG-5 and MKG-5 samples showed limited distributions and volumes of the gel pores (Figure 2c,d).

### 3.2. Strength

Figure 3 shows the statistical compressive strengths of the MKG mortars. Clearly, all the mortars showed relatively high strengths (>57 MPa), suggesting that the mixes used in this work could fabricate geopolymer materials with good mechanical properties. Similar strength data were reported elsewhere [11,17].

Most granular-compacted materials (like concrete and geopolymer materials) accumulate strengths by increasing the compactness of solids or decreasing the volume of pores. Therefore, total porosity can be an indicator to predict material strength. Figure 4 shows the plots of compressive strength against total porosity. Roughly, the compressive strength decreased linearly with the total porosity, which can be expressed as:(2)$$\sigma =\sigma_{0}\left(1-\alpha \phi \right)$$
where σ and $\sigma_{0}$ are the compressive strengths at current porosity ϕ and zero porosity, respectively; α is the coefficient. Equation (Equation 2) is also known as the Hasselmann equation for the strength-porosity relationship of porous materials [43].

Fitting of the compressive strength data with Equation (Equation 2) yielded $\sigma_{0}=99.74$ MPa, α=0.022. The obtained $\sigma_{0}$ was much larger than that of alkali-activated slag mortars [44], but close to that of geopolymer mortars prepared by slag, fly ash, and palm oil fuel ash [18]. The coefficient α was much lower than that reported in the literature [18,44]. The results implied the high strength of the solid matrix and the limited influence of total porosity.

### 3.3. Roles of the Materials’ Composition

In order to better understand the effects of the Si/Al ratio and the Na/Al ratio on the strength and pore structure of the MKG mortars, specific analyses were performed on each factor. Figure 5 shows the compressive strengths and pore volumes in terms of the Si/Al ratio and the Na/Al ratio. The mix with the Si/Al ratio at 2.32 showed the highest compressive strength (σ=66.04 MPa), while the mixes with the other Si/Al ratios showed slightly lower compressive strengths (Figure 5a). Meanwhile, increasing the Si/Al ratio promoted the capillary pore volume, but decreased the gel pore volume (Figure 5c).

When the Si/Al ratio was fixed at 2.62, the changes of the Na/Al ratio caused monotonous changes of both compressive strengths and pore volumes. For example, as the Na/Al ratio increased from 1.01 to 1.36, the compressive strength increased from 57.63 MPa to 74.28 MPa with the promotion extents of 29%, and the gel pore volume decreased from 0.0365 mL/g to 0.0093 mL/g with the decreasing extents of 75% (Figure 5b,d). The absolute capillary volume only showed minor changes as the Na/Al ratio varied (Figure 5d), but the relative ratio of capillary pores increased significantly (Figure 2d).

The changes of strength and pore structure were the consequences of physico-chemical interactions among the metakaolin particles and alkaline activator. Because both the Si/Al ratio and the Na/Al ratio were optimally selected according to the literature data [17] and our previous work [19], all the MKG mortars showed high compressive strengths. Our data did not follow the observation that materials with higher capillary pore volumes have lower strengths [45]. The reason is that the enhanced polymerization extents of geopolymer at the higher Na/Al ratio (or Na/Si ratio) could increase the compactness of the nanostructure, thus decreasing the nanopores and promoting the capillary pores [20].

## 4. Freezing-Thawing Damages

### 4.1. Morphology, Mass Loss, and Strength Loss

Figure 6 shows the photos of the MKG mortars after 50 F-T cycles. Serious surface spalling occurred on the MKG-1 specimens (Figure 6a), causing a mass loss of 21.04% (Figure 7a). The MKG-2 and MKG-3 specimens were completely damaged or pulverized into small mortar particles (Figure 6b,c), thus referred to as 100% mass loss (Figure 7a). The MKG-4 and MKG-5 specimens showed minor or negligible visual changes (Figure 6d,e) and mass loss (Figure 7a).

Owing to the severe spalling and pulverization, the specimens of MKG-1, MKG-2, and MKG-3 were not suitable for strength tests. Those mortar mixes were therefore regarded to have 100% strength loss after F-T loads (Figure 7b). While the MKG-4 and MKG-5 mortars showed minor surface spalling, they had severe strength losses, i.e., 10.18% for MKG-4 and 36.74% for MKG-5 (Figure 7b). This implied that serious F-T damages occurred in the materials. The results obtained from Figure 6 and Figure 7 suggested that the order of F-T resistance of the MKG mortars could be rated as: MKG-4 > MKG-5 > MKG-1 > MKG-2 = MKG-3. This rate was roughly in accordance with the strength and porosity data provided in Section 3: materials with a higher strength (or lower porosity) also had better F-T resistance.

SEM/BSE tests were conducted on the MKG-4 and MKG-5 specimens with complete appearance to explore the F-T damages. Figure 8 selectively illustrates the BSE images of the MKG-4 and MKG-5 mortars. Clearly, several tortuous microcracks can be observed in the materials, which connected some air voids and/or big pores. Those cracks were different from the drying cracks that should be distributed in the geopolymer matrix with smeared cracking meshes [46]. The cracks would greatly decrease the compressive strength, but maintain the specimen mass before they were maturely developed to form the large cracks that caused the material spalling. While the MKG-5 specimen showed less cracks than the MKG-4 specimen, the cracks in the former material connected with each other, forming percolated cracks. This explained why the MKG-5 mortar showed severer mass and strength losses (Figure 7).

### 4.2. Pore Structure Alterations

The pore structure alterations of the MKG mortars after F-T loads were assessed by MIP tests. Figure 9 shows the PSD spectra and pore segmentation of the MKG mortars after F-T loads. At a first glance, both the APSD and DPSD spectra showed similar curves to those before F-T loads. This was reasonable, because F-T loads would not change the spatial compactness of the material system. For example, the first mercury rise in APSD and the peak in DPSD remained unchanged after F-T loads for most mixes, except that a slight rise in this part occurred for the MKG-5 mortar (Figure 9e). The F-T cracking enhanced the surface roughness, which, consequently, raised the surface-conformance effect (Figure 9f).

Careful examinations on two phases of the DPSD spectra were performed, which helped characterize the pore structure alterations caused by F-T loads. Phases A and B represent the gel pores below 50 nm and the capillary pores between 100 and 10,000 nm, respectively (Figure 9b). Comparative plots of the two phases between the mortars before and after F-T loads are drawn in Figure 9c,d. Surprisingly, the gel pores of all mortar mixes were raised (Figure 9c). For instance, the MKG-2 and MKG-3 mixes that appeared to have the severest F-T damages (Figure 6) showed the heaviest rises of the gel pores, and the gel pore ratios increased from 39% and 37% (Figure 2d) up to 44% and 43% (Figure 9f), respectively. Meanwhile, the capillary pores were narrowed (Figure 9d), and the pore volumes were decreased (Figure 9e). Those were contrary to the pore structure degradation reported elsewhere [23]. The anomalous gel and capillary pores’ alterations did not directly reflect the material degradation caused by F-T loads, but might indicate the pore refinement induced by the continual polymerization of the materials immersed in water. A literature survey indicated that the first few F-T cycles may enhance the geopolymerization process, which helps refine the pore structure and even increase the strength [47,48,49]. The products of geopolymerization filled the coarse pores, which thus decreased the capillary pore fractions of MKG-4 and MKG-5 shown in Figure 9f.

### 4.3. Further Discussion: Permeability Associated Pressure Relaxation

In this work, experimental tests on the MKG mortars verified the compositional sensitivity of the strength and pore structure. Increasing the Na/Al ratio tended to enhance the material strengths. A linear relationship was found between the compressive strength and total porosity (Figure 4). Except the volumes induced by the surface-conformance effect showing similar results, both the capillary pores and gel pores were highly dependent on the Si/Al ratio and the Na/Al ratio (Figure 5). While our experimental data testified that the geopolymer material with a higher strength also showed a better F-T resistance, we also observed the anomalies of the pore structure alterations after F-T loads (Figure 9). These data also raised a question: Why does an MKG material with a coarser pore structure show better F-T resistance? To address this question, we should understand the roles different pores played during freezing.

Figure 10 schematically illustrates the snapshots of a thin pore system and a coarse one during freezing. In the thin pore (or coarse pore) system, a big pore chamber was connected by thin channels (or coarse channels), and all the pores were filled with water before freezing. The thin pore system may capture the pore structures of MKG-1, MKG-2, and MKG-3, and the coarse one may represent those of MKG-4 and MKG-5; see Figure 2.

When temperature decreased to a subzero value, the water confined in the big pore chamber began to freeze while the pore channels remained unfrozen. Freezing pressures (including the hydraulic pressure and crystallization pressure [33]) caused by the phase transition of water to ice in the confinement built up. The crystallization pressure would be locally exerted on the pore walls due to the free-energy differences between water and ice [32]. The hydraulic pressure that was exerted homogeneously on the pore walls through the unfrozen pore water, however, could be relaxed through the viscous flow of the pore water (Figure 10). This important feature was used by concrete scientists who homogeneously entrained air voids in the material matrix to shorten the water flow distance and thus mitigate the hydraulic pressures [50].

Generally, the hydraulic pressure relaxation is inversely proportional to the permeability of pore channels for water flow, which can be expressed as [51]:(3)$$\Delta P_{h}\propto Lt/K_{h}$$
where $\Delta P_{h}$ is the changes of hydraulic pressure by relaxation, *L* is the pore length, *t* is the relaxation time, and $K_{h}$ is the permeability. According to Katz and Thompson [40], the permeability of a porous material can be predicted by the threshold pore size, $K_{h}\propto D_{T}^{-2}$. Equation (Equation 3) can be rewritten as:(4)$$\Delta P_{h}\propto D_{T}^{2}Lt$$
or the pore pressure relaxation time *t* can be expressed as:(5)$$t\propto \Delta P_{h}/LD_{T}^{2}$$

From Equations (Equation 4) and (Equation 5), it is easy to understand that less pore pressure relaxation can occur in the thin-pore system with the same relaxation time, or more time is required to relax the same pore pressures. Let us take the pore structure data of MKG-2 and MKG-5 as examples, and assume that the ice formation in the big pores accumulated the hydraulic pressures up to 10 MPa, while the pore length *L* was the same. The MKG-2 mortar required ${\left(1049.4/350.1\right)}^{2}\approx 9$ times longer than MKG-5 to relax the hydraulic pressure at 10 MPa according to Equation (Equation 5). This may explain why the MKG materials with the coarser pore structures showed better F-T resistances.

As the last comment, we should understand that the experimental data provided in this work did not imply that a material with a coarser pore structure would have better durability. Indeed, to promote the F-T resistance, the homogeneous entertainment of air voids in the material matrix may be the most effective routine. As shown in Figure 8, the MKG mortars with high F-T resistances (i.e., MKG-4 and MKG-5) contained air voids to accommodate the expelled water and mitigate the hydraulic pressures built by ice formation in the pores. Furthermore, the coarser pores would greatly decrease the material permeability, which in turn would decrease the resistance against the transport of harmful ions in the materials. Further rigorous pore structure design is required to optimize the properties of MKG materials before engineering applications.

## 5. Conclusions

The MKG mortars with different Si/Al ratios and Na/Al ratios showed different MIP pore structures. Except the first mercury rises, which were identical to the surface fillings of the mercury under low pressures showing negligible volume changes, both the capillary pores and gel pores were greatly impacted by the Si/Al ratio and the Na/Al ratio. Increasing both the Si/Al ratio and the Na/Al ratio decreased the gel pores, but promoted the capillary pores.All the MKG mortars showed relatively high strengths. The strength decreased with increasing the total porosity, which roughly followed a linear plot.The MKG mortars showed different F-T resistances: MKG-4 > MKG-5 > MKG-1 > MKG-2 = MKG-3. The MKG mortars (MKG-1, MKG-2, and MKG-3) at Na/Al ratios lower than 1.26 all showed serious F-T damages. Increasing the Na/Al ratio promoted the F-T resistance. F-T loads also caused obvious cracking of MKG-4 and MKG-5.MIP tests showed that the pore structures were refined after F-T loads, which was probably caused by the continual curing of the materials. Materials with finer pore structures showed worse F-T resistances owing to the slower pore pressure relaxation rates.

## Figures and Tables

**Figure 1 materials-13-02973-f001:**
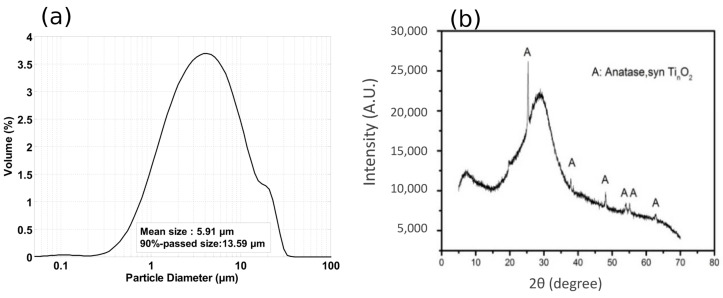
Particle size distribution (**a**) and XRD patterns (**b**) of the metakaolin powder.

**Figure 2 materials-13-02973-f002:**
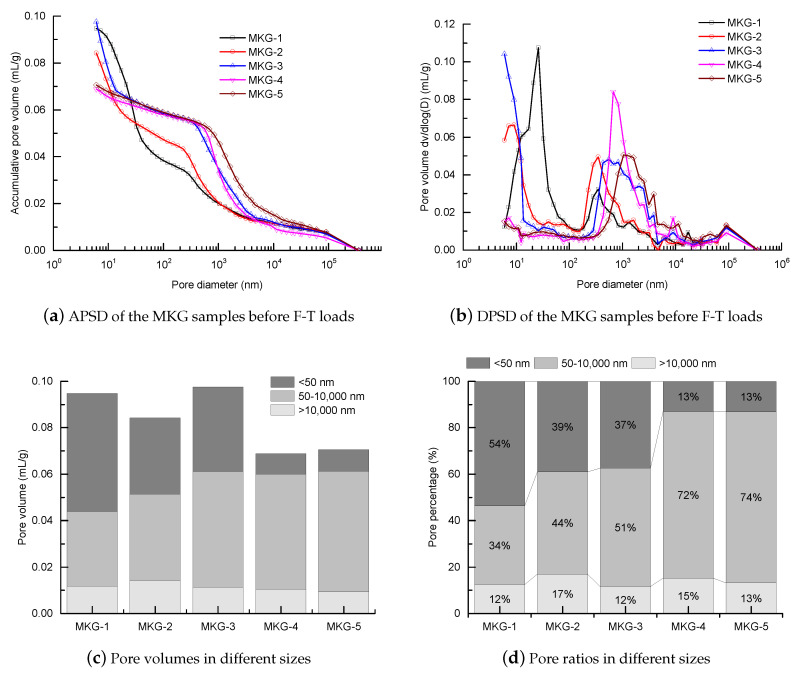
Pore structure of the MKG mortars before freezing-thawing (F-T) loads: (**a**) accumulative pore size distribution (APSD) spectra; (**b**) differential PSD (DPSD) spectra; (**c**) pore volumes in different sizes; and (**d**) pore ratios in different sizes.

**Figure 3 materials-13-02973-f003:**
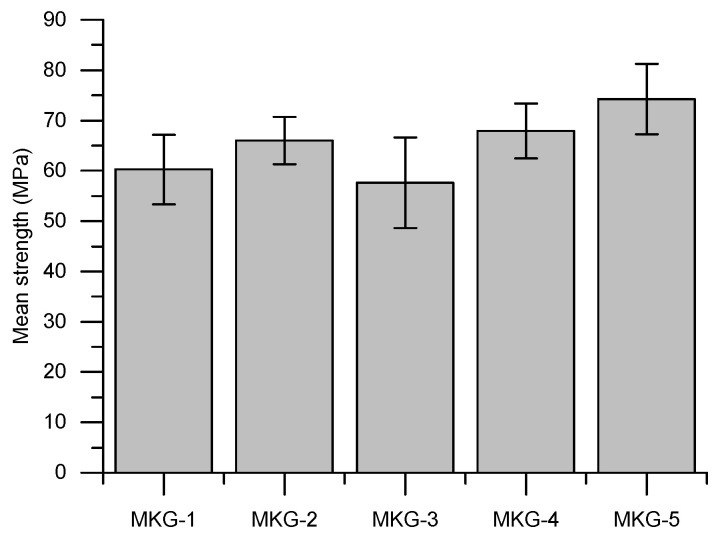
Statistical strengths of the MKG mortars.

**Figure 4 materials-13-02973-f004:**
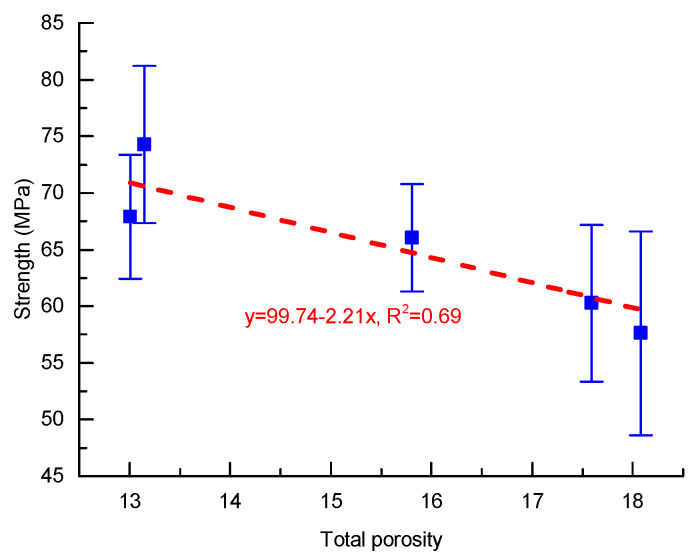
Relationships between strength and porosity.

**Figure 5 materials-13-02973-f005:**
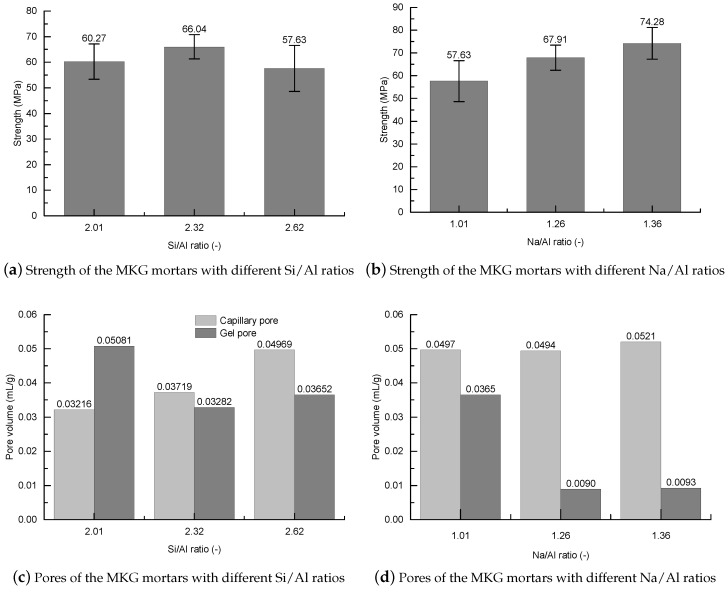
Compositional dependence of the strength and pore structure of the MKG mortars: strength of the MKG mortars versus (**a**) the Si/Al ratio; (**b**) Na/Al ratio, pores of the MKG mortars versus the (**c**) Si/Al ratio and (**d**) Na/Al ratio.

**Figure 6 materials-13-02973-f006:**
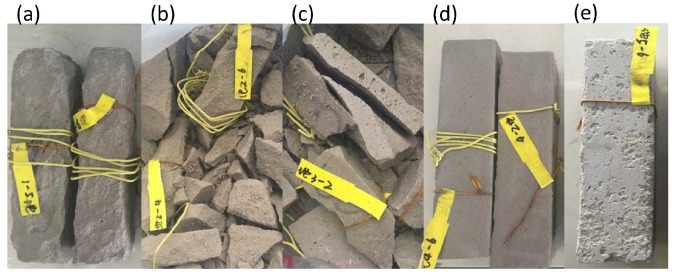
Pictures of the MKG mortars after 50 F-T cycles: (**a**) MKG-1; (**b**) MKG-2; (**c**) MKG-3; (**d**) MKG-4; and (**e**) MKG-5.

**Figure 7 materials-13-02973-f007:**
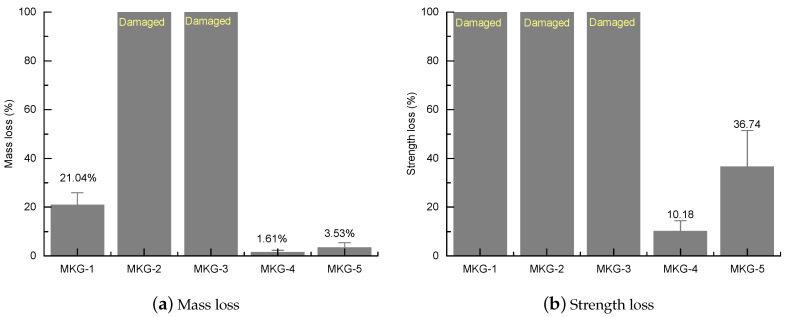
Mass loss (**a**) and strength loss (**b**) of the MKG mortars after 50 F-T cycles.

**Figure 8 materials-13-02973-f008:**
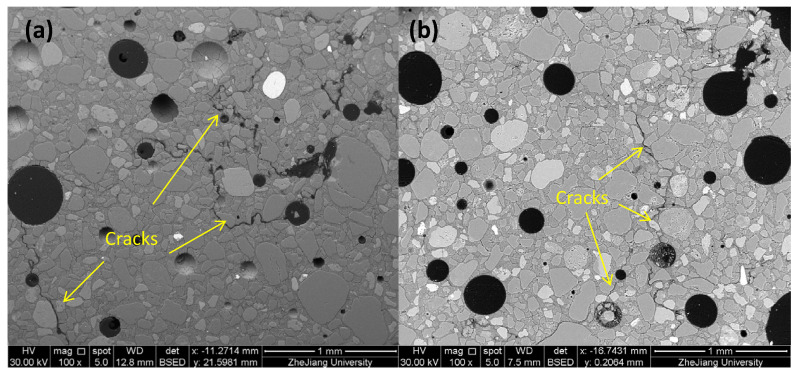
Typical SEM/BSE pictures of (**a**) MKG-4 and (**b**) MKG-5 specimens after F-T loads.

**Figure 9 materials-13-02973-f009:**
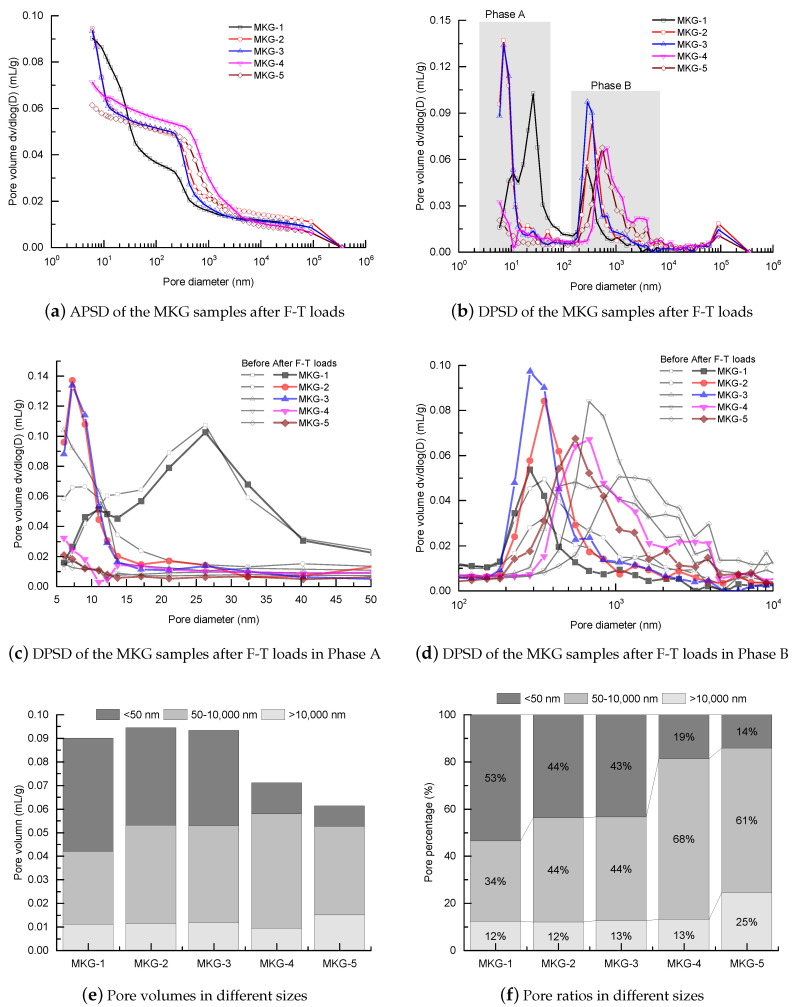
Pore structure of the MKG mortars after F-T loads: (**a**) APSD spectra; (**b**) DPSD spectra; comparison of the pores in Phase A (**c**) and Phase B (**d**); (**e**) pore volumes in different sizes; and (**f**) pore ratios in different sizes.

**Figure 10 materials-13-02973-f010:**
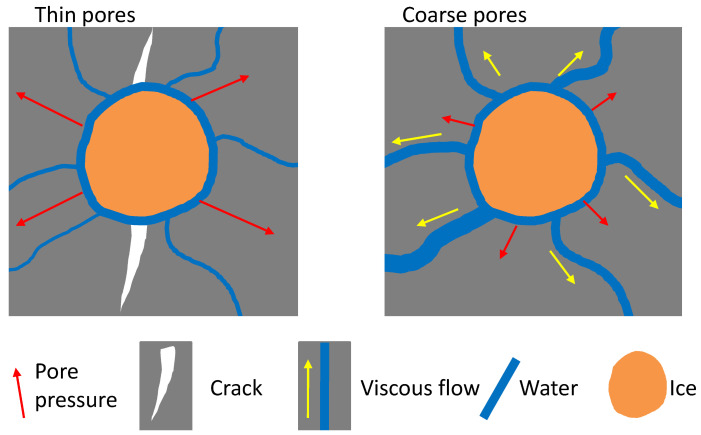
Schematic illustration of the freezing resistance differences between a material with thin pores and that with coarse pores.

**Table 1 materials-13-02973-t001:** Chemical components of the metakaolin powder.

Composition	${\mathrm{SiO}}_{2}$	${\mathrm{Al}}_{2}O_{3}$	${\mathrm{TiO}}_{2}$	${\mathrm{Fe}}_{2}O_{3}$	${\mathrm{Na}}_{2}O$	$K_{2}O$	CaO
Mass content (%)	57.47	39.81	1.79	0.43	0.27	0.21	0.04

**Table 2 materials-13-02973-t002:** Mix proportions of the metakaolin-based geopolymer (MKG) concretes.

Mix ID	Metakaolin	Water Glass	NaOH	Water	Si/Al	Na/Al
MKG-1	1016	640	193	410	2.01	1.01
MKG-2	936	993	138	197	2.32	1.01
MKG-3	868	1290	84	17	2.62	1.01
MKG-4	842	1251	136	31	2.62	1.26
MKG-5	832	1233	156	38	2.62	1.36

**Table 3 materials-13-02973-t003:** Characteristic pore parameters of the MKG mortars form mercury intrusion porosimetry (MIP) tests.

Sample	Total Porosity (%)	Average Pore Size (nm)	Specific Surface Area ($m^{2}/g$)	Threshold Pore Size (nm)
MKG-1	17.59	32.0	11.83	350.1
MKG-2	15.81	27.1	12.43	350.1
MKG-3	18.08	25.2	15.46	553.7
MKG-4	13.01	84.5	3.25	675.9
MKG-5	13.15	89.9	3.14	1049.4

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
