# Peer review of "Compositional Dependence of Pore Structure, Strengthand Freezing-Thawing Resistance of Metakaolin-Based Geopolymers"

_materials, 2020, doi:10.3390/ma13132973_

Round 1
Reviewer 1 Report
Dear Authors,
Thank you for submitting this paper on metakaolin-based geopolymers. There is still little information on freeze-thaw behaviour of geoploymers therefore I am glad to be able to review this manuscript.
I have two concerns which are effecting each other. Firstly, the written English is good but in some places a little bit sloppy, secondly as there is not so much information about this topic published it would be great to state throughout the paper, why you choose exact these parameters. One example is that why you choose the temperatures of -17 and 8 deg. Celsius for the F-T tests. In this special issue you and all other authors are ambassadors for this new material, so try to state exactly why you choose to measure these properties and why in exact this way, so that your paper contributes to useful knowledge in the building materials community.
Author Response
Please find the attached file for the responses to the reviewer 1.

Reviewer 2 Report
In this paper the Authors investigate the influence of Si/Al and Na/Al ratios on the physical and mechanical characteristics of metakaolin-based geopolymer (MKG) mortars. Specifically, different mortar samples were fabricated by varying the Si/Al and Na/Al ratio: compressive strengths, freezing-thawing (F-T) resistance, pore structure and micro-morphology were thoroughly investigated. From the results, the Authors show that different compositions lead to different pore structures, especially in terms of capillary and gel pore volumes. In turn, this is correlated with the different compressive strengths registered for the different specimens. F-T analyses also showed that the Si/Al and Na/Al ratios play an important role in defining the F-T resistance due to the different pore pressure relaxation rates.
The manuscript is very interesting and fits well with the aim of the “Materials” Journal. Moreover, it is very well written and the results are presented in a clear and organized manner. For this reason, my opinion is that it can be accepted for publication.
Just as a minor suggestion to increase the interest of the paper to a broader readership, the Authors are encouraged to expand the literature survey about the freezing-thawing phenomenon and the consequent resistance after F-T cycles of other structural materials, such as concrete, mortar and masonry.
Among others, the Authors can refer to the following papers:
- P. Bocca, A. Grazzini (2013) Mechanical properties and freeze-thaw durability of strengthening mortars. ASCE Journal of Materials in Civil Engineering, 25, 274-280.
- A. Carpinteri, A. Grazzini, G. Lacidogna, A. Manuello (2014) Durability evaluation of reinforced masonry by fatigue tests and acoustic emission technique. Structural Control and Health Monitoring, 21, 950-961.
- I.F. Saèz del Bosque et al. (2020) Freeze-thaw resistance of concrete containing mixed aggregate and construction and demolition waste-additioned cement in water and de-icing salts. Construction and Building Materials, 259, 119772.
According to what said above, the reviewer’s opinion is that the manuscript can be accepted for publication after the described minor revisions.
Author Response
Please find the attached file for the responses to the reviewer 2.

Reviewer 3 Report
According to my opinion, It is an interesting manuscript and I consider that it should be accepted after minor revision:
a) The XRD characterization process should be described (method, instrument, ect.)
b) All the materials (MKG1-MKG5) (not only the commercial material) should be characterized with the XRD method. It would be interested to see if, what kind and why there is a difference among the materials.
c) Minor corrections on the Reference list should be done (e.g., [2], [5], [12])
Author Response
Please find the attached file for the responses to the reviewer 3.

Reviewer 4 Report
In this manuscript, the correlations between metakaolin-based geopolymers formulation (in particular Si/Al and Na/Al ratios) and freeze-thaw resistance were investigated. The manuscript is well written, is scientifically sound and the overall merit is the systematic analysis of the influence of Si/Al and Na/Al molar ratios on the MK-based geopolymer microstructure and their influence on freeze-thaw resistance. However, some corrections and additions are necessary to improve manuscript clarity and results discussion.
1) Abstract: more details about investigated compositions and the main results are necessary (molar ratios, which is the best mortar in terms of F-T resistance, pores microstructure etc.);
2) Introduction must be improved: Concerning sustainability of geopolymers and alkali-activated materials, a debate among researchers is ongoing but the main advantage in terms of sustainability is represented by the use of waste or by-products as raw materials in the preparation of binders alternative to Portland Cement. Metakaolin is not the most sustainable material in these terms, because kaolin become "reactive" only after calcination (resulting in MK). Please, revise lines 23-25, commenting more recent articles about sustainability of geopolymers and alkali-activated materials and discussing about the use of waste materials such as mineral wastes, construction and demolition wastes, metallurgical slag etc. (e.g. Assi et al., https://doi.org/10.1016/j.jclepro.2020.121477; El-Gamal and Selim, https://doi.org/10.1016/j.susmat.2017.03.001; Bassani et al., https://doi.org/10.1016/j.jclepro.2019.06.207).
Finally, a state of the art concerning freeze-thaw resistance of geopolymers (in particular MK-based) must be briefly discussed to better understand the literature gap and the novelty of the work (only one sentence is present, lines 52-53).
3) Section 2.1: line 69: revise "utmost mass"; Table 1 remove the third decimal to the SiO2 mass content; XRD pattern of Figure 1 is not commented (amorphous halo, anatase presence etc.); Figure 1 caption: XRD patterns; the y axis title of Figure 1b is missing; Table 2 units are missing (wt%? mass? volume? molar ratios? etc.) Did authors measured or assessed mortars viscosity? Viscosity is important during samples casting and preparation as directly influences mortars microstructure.
4) Section 2.2: Please, clarify and revise the following sentence (lines 105-106): Those vessels that contained the MKG specimens and water were then removed into the F-T testing machine. Generally ASTM (ASTM C666 / C666M - 15) or European standards (EN 14617-5:2012) are used for F-T test, why authors decided to perform a different test?
5) Results and discussion: As expected, MK-based geopolymers properties are strictly correlated to mortars pore structure, as determined via Mercury Intrusion Porosimetry (MIP). Indeed, authors based all the discussion and their conclusions on this test. However, at least three measurements are required to assess if a result is reproducible or not, particularly with a test like MIP that can be not representative since performed on a very small portion of sample. Please, provide a standard deviation for MIP results (Table 3, Figures 2 and 9). Moreover, the legend of Figure 2a-b is different from that of Figure 2c-d. Please, add samples designation (MKG-1 etc.) also to APSD and DPSD curves. In Figure 8, authors reported the micrographs of only two samples. However, it could be useful to compare also the cracking occurring in a damaged sample. Did authors performed also a statystical analysis of cracking phenomena (such as cracks width, lenght and numbers)? Why MKG-4 and 5 behave differently even if the starting porosity is the same (both in terms of total porosity and pores size)? Only slight differences are recognizable among porosimetry results before and after freeze-thaw tests (Fig 2 and Fig 9, respectively). The biggest difference is observed for MKG-5. Why pores refinement was higher in this case? The higher Na amount increased the alkalinity improving metakaolin dissolution even at later ages?
Author Response
Please find the attached file for the responses to the reviewer 4.
